# Ultrafast current imaging by Bayesian inversion

S. Somnath [1,2], K.J.H. Law[1,3], A.N. Morozovska [4], P. Maksymovych[1,2], Y. Kim[5], X. Lu [6], M. Alexe[7], R. Archibald[1,3], S.V. Kalinin [1,2], S. Jesse[1,2] & R.K. Vasudevan [1,2]

Spectroscopic measurements of current–voltage curves in scanning probe microscopy is the earliest and one of the most common methods for characterizing local energy-dependent electronic properties, providing insight into superconductive, semiconductor, and memristive behaviors. However, the quasistatic nature of these measurements renders them extremely slow. Here, we demonstrate a fundamentally new approach for dynamic spectroscopic current imaging via full information capture and Bayesian inference. This general-mode $I–V$ method allows three orders of magnitude faster measurement rates than presently possible. The technique is demonstrated by acquiring $I–V$ curves in ferroelectric nanocapacitors, yielding >100,000 $I–V$ curves in <20 min. This allows detection of switching currents in the nanoscale capacitors, as well as determination of the dielectric constant. These experiments show the potential for the use of full information capture and Bayesian inference toward extracting physics from rapid $I–V$ measurements, and can be used for transport measurements in both atomic force and scanning tunneling microscopy.

[1] Center for Nanophase Materials Sciences, Oak Ridge National Laboratory, Oak Ridge, TN 37831, USA. [2] Institute for Functional Imaging of Materials, Oak Ridge National Laboratory, Oak Ridge, TN 37831, USA. [3] Computer Science and Mathematics Division, Oak Ridge National Laboratory, Oak Ridge, TN 37831, USA. [4] Institute of Physics, National Academy of Sciences of Ukraine, 46, pr. Nauky, Kyiv 03028, Ukraine. [5] School of Advanced Materials Science and Engineering, Sungkyunkwan University (SKKU), Suwon 16419, Republic of Korea. [6] The State Key Discipline Laboratory of Wide Band Gap Semiconductor Technology, Xidian University, Xi'an, 710071 Shaanxi, China. [7] Department of Physics, University of Warwick, Coventry CV4 7AL, UK. Correspondence and requests for materials should be addressed to R.K.V. (email: vasudevanrk@ornl.gov)

Since its emergence in early 1980s, scanning probe microscopy (SPM) has become one of the most powerful and versatile methods in the nanotechnology and condensed matter physics toolkits, enabling functional imaging, manipulation, and characterization at atomic and even sub-atomic resolution. Since its inception, it was recognized that SPM can offer a diverse array of spectroscopic imaging techniques and coupled modalities (see also, https://journals.aps.org/prl/scanning-probe-microscopy)[1–3]. The key feature of these is the ability to spatially correlate behaviors, based on a spectroscopic experiment performed on a grid of points, which can be correlated with another channel of information (e.g., topography) in the same region. Typically, these spectroscopic techniques are slow (~minutes to hours), and thus require stable instrumentation platforms free from drift, as well as samples that are not prone to degradation. Most importantly, capturing transient phenomena requires ingenious approaches[4–6], which are typically not transferrable across different types of measurements. These considerations both guided the development of ultrastable SPM platforms and low noise facilities, and limited the spectrum of materials and hence fundamental physical phenomena that could be explored.

To overcome these restrictions, much attention has been focused on increasing the acquisition rates within a scanning probe measurement. For example, techniques to perform a force–distance curve[7] (in atomic force microscopy (AFM)) in a rapid fashion are now commercially available[8]. Similar strides have been made in fast imaging, including fast imaging of biomolecular processes[9,10] as well as nucleation and growth of ferroelectric domains[11,12]. Indeed, the use of small cantilevers with high resonance frequencies enabled the development of video-rate scanning in AFM. Ginger et al.[13,14] have developed a method allowing for probing transient phenomena with 10–100 ns temporal resolution in non-contact mode. Recently, a subset of the current authors developed a technique involving full information capture, enabling both fast Kelvin Probe Force Microscopy measurements[15] as well as the ability to acquire ferroelectric strain (hysteresis) loops at three orders of magnitude faster rates[16]. However, the most basic spectroscopic method, where the probe (sample) is biased and current through the sample (probe) is collected, still suffers from a substantial time cost, the limitation stemming from the quasistatic nature of the current–voltage (I–V) measurements.

The I–V measurements are perhaps the oldest 'spectroscopic' mode in SPM, having been referenced in the original works in the 1980s on the birth of scanning tunneling microscopy (STM)[17,18]. The technique has allowed an enormous number of electronic phenomena to be measured in a spatially resolved fashion and correlated to individual features of the sample, such as grain boundaries[19], domain walls[20], phase boundaries[21], and in the case of STM, individual atoms and molecules[22,23]. For example, in STM, the use of $dI/dV$ maps allows for the investigation of the local density of states[24], as well as exploration of phenomena such as quasiparticle scattering[25] and superconducting gaps[26,27], dramatically increasing our knowledge of electronic structures at the nanoscale and providing the experimental counterpart to electronic band structure calculations from first-principles methods. Thus, enhancing the speed of acquisition is of critical importance to enable further advances in a multitude of fields within nanoscience.

In this communication, we report a fundamentally new approach to the problem, based on the combination of full information capture from the current amplifier, AC excitation (as opposed to standard DC voltage waveforms), and Bayesian inference in order to recover I–V curves at rates hundreds of times faster than the current state of the art. We term this method 'general-mode I–V' (G-IV) acquisition, and show its utility via detection of switching currents in ferroelectric nanocapacitors, and compare it with standard IV acquisition on the same area. In addition to increased spatial resolution, the method also provides information on the tip–sample capacitance, which allows determination of the dielectric constant within each nanocapacitor. Analysis of individual switching currents enables determination of disorder in each nanocapacitor, providing a complementary route to existing piezoresponse force microscopy-based schemes. The method in general can be readily applied in both AFM and STM, and further allows access to current measurements as a function of frequency in the 10–100s Hz range, opening new avenues for nanoscale electronic measurements.

## Results

**I–V considerations**. The excitation waveform in standard I–V spectroscopy (henceforth referred to as S-IV) is typically a sequence of DC pulses that follow an envelope of a triangular waveform, at a low frequency (a few Hz at most), which is applied to either the tip or the sample (see Fig. 1a, b). This causes a current (tunneling in STM) which can be measured either through the tip or the sample, which is of the form

$$I = \frac{V}{R} + C\frac{dV}{dt},$$ (1)

where $V$ is the voltage applied to the tip or sample, and $R$ and $C$ are the resistance and capacitance of the circuit, respectively. Though this is the typically used expression, we found through calibration experiments (detailed in Supplementary Methods and Supplementary Figures 1,2) that the current is better modeled by

$$I = V\left(\frac{1}{R} + a_0 C\right) + C\frac{dV}{dt},$$ (2)

where $a_0$ is a constant on the order of $(50–300)$ $F^{-1}\Omega^{-1}$ and depends on the experimental setup. For the following, we assume that the constant $a_0$ is known. To minimize the contribution to the current from the second term in Eq. (1), the measurement step includes a delay time $t_1$, and minimization of noise requires integration for some finite time $t_2$ so that the measured current $I_j$ at voltage $V_j$ is given as

$$I_j = \frac{1}{t_2 - t_1}\int_{t_1}^{t_2} I(t, V_j)dt.$$ (3)

For most setups, $t_1$ and $t_2$ are typically in the hundreds of μs to several ms range, such that obtaining reasonable spectral (voltage) resolution in the experiment requires slowly varying waveforms. Given that the currents are also extremely small, ranging from ~fA to μA, $I(t)$ is amplified by the use of a current amplifier by a gain on the order of ~$10^6$–$10^9$, allowing detection of currents with noise floors of ~500 fA in commercial SPM systems. This inevitably brings in the issue of the parasitic capacitance, which is normally large enough to obscure current measurements even at frequencies of ~10 Hz. The current amplifiers, however, have a bandwidth of 1–2 kHz—potentially enabling two to three orders of magnitude faster acquisition times. Note that lock-in amplifier can decouple capacitive and resistive circuits at any achievable frequency. However, the decoupling comes at the cost of comparatively long integration time, and therefore, again limited overall measurement speed.

The dynamic IV measurement mode relies on use of Bayesian methods to infer the capacitive and resistive contributions. We analytically have considered the case where the measurement system consists of a non-linear resistor and a capacitor in series (one case) and in parallel (another case) (see Supplementary

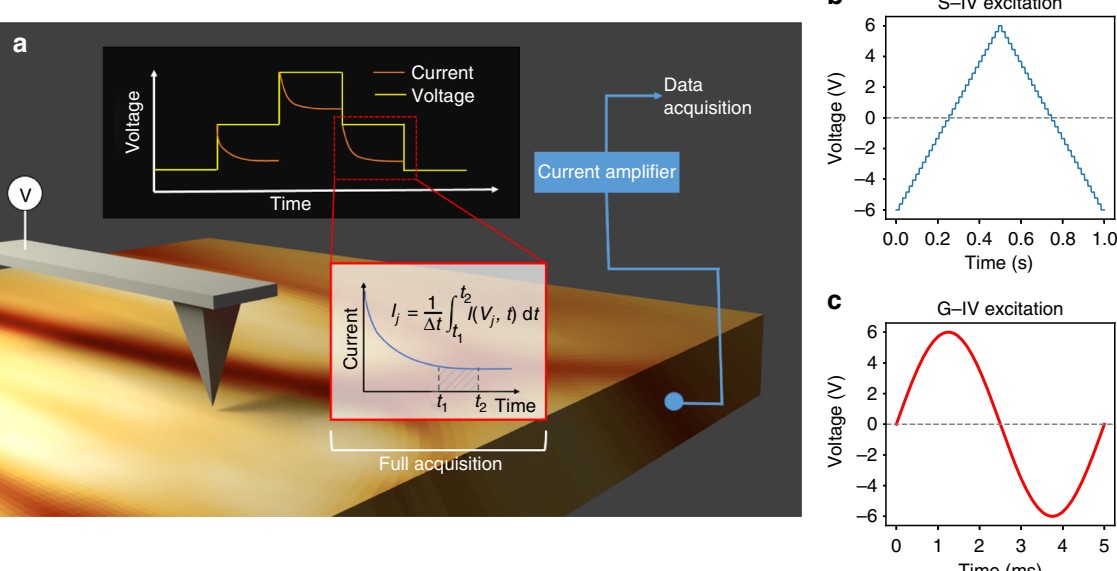

**Fig. 1** Schematic for *I–V* measurements in contact-mode AFM. **a** In this schematic, the tip is biased with respect to the bottom electrode which is grounded. Current flows are detected and amplified by use of a high-gain current amplifier, with the output fed into a data acquisition system. **b** In an S-IV measurement, a slowly varying DC waveform is applied to the tip (or sample) and the current is measured; each step in the voltage results in a relaxation in the measured current due to the capacitance in the circuit. By neglecting some portion of this current and averaging over some integration time $\Delta t$, a single value of the current $I_j$ at $V_j$ is recorded. In general-mode *I–V*, the waveform is fast (see **c**) (generally, sinusoidal but not limited to this case), and the full current stream is recorded, allowing filtering after collection as appropriate

Note 1 for analysis and Supplementary Figures 3–5 for simulation). Given that the AFM current measurement system has a capacitance in parallel with the voltage circuit, we may consider that Eq. (2) is generally valid, and the task of determining $R(V)$ falls on the ability to determine $I(V)$ and $C$, assuming $C$ is constant in the measured voltage range (this assumption can be tested by experiments). It should be noted that capacitance variation across the sample will add to the capacitive contribution, and so $C$ need not be spatially homogeneous, i.e., $C = C(x,y)$. With $C$ unknown and in the presence of noise in $I$, the direct inversion of Eq. (1) to find $R(V)$ necessitates alternative methods.

One method that can be utilized to infer $C$ and $R(V)$ is Bayesian inference, which has been used in e.g., crystal structure refinement[28,29] and for analysis of first-principles data[30]. The Bayesian framework enables one to seamlessly incorporate prior knowledge and expertise with physical models and noisy data in order to reconstruct quantities of interest and their associated uncertainty[31,32]. Here, the main objective is to reconstruct most probable values of $R(V)$ and $C$ which respect the model and data, in the sense that the error of the reconstructed signal with respect to the measured current is small, along with an estimation of the uncertainty. Leveraging the simple observation that the current is linear in $S = R^{-1}$ and $C$ enables the design of a valid simple linear Gaussian statistical model in which the posterior $P(S,C|y)$ can be computed quickly in closed form. Subsequently, valid estimators for $R$ and $C$ can be constructed by post-processing to remove unphysical artifacts arising from lacking information.

**General-mode *I–V*.** Our technique, which we term general-mode *I–V* (G-IV), is based on a family of recently developed imaging and spectroscopy techniques where the complete response from the sensor is acquired and retained, as opposed to averaging the response over a time window (in the case of conventional *I–V* techniques) or retaining the signal from a single frequency (heterodyne detection methods such as lock-in amplifiers or phase-

locked loops)[33–35]. In G-IV, the tip is biased by a fast sinusoidal (100–500 Hz) waveform and the raw signal from the current amplifier is acquired (Fig. 1c). Importantly, G-IV curves can be acquired with high spatial resolution compared to the conventional technique since each G-IV curve is acquired within a few milliseconds (i.e., comparable to a single pixel time at standard imaging rates) instead of seconds as in the conventional techniques, enabling fast imaging and obviating effects of drift and sample instabilities.

We note here that there is a price to pay for the fast measurements—inherently the faster measurements will suffer from lower signal to noise, and also introduce more uncertainty in the measurement due to the nature of the inference. However, appropriate de-noising using Fourier filtering appears to work reasonably well, at the cost of very fine features in the current signal (if these are present). At the same time, the nature of the inference allows us to determine the bounds, as we achieve valid statistical estimates of the covariance using this technique. In other words, we trade signal to noise for substantially fast measurements with higher (but known) uncertainty.

To demonstrate the dynamic G-IV measurements, we use a model system of a prototypical ferroelectric (001) $PbZr_{0.2}Ti_{0.8}O_3$ (PZT) sample of thickness 90 nm, deposited via pulsed laser deposition on $SrRuO_3/SrTiO_3$ substrates. Au/Cu pads of nominal size ~380 nm were fabricated via an anodic aluminum oxide-based fabrication method completing the nanocapacitor structure (see ref. 36 for more details). Given the technological importance of ferroelectric materials, including their use in actuators, sensors, energy harvesters, and random access memories[37], it is desirable to obtain information for nanoscale systems from a complementary channel that extends beyond the traditional piezoresponse force microscopy-based techniques[38]. In this regard, analysis of the displacement current (occurring from polarization reversal) can be highly useful[39]. We attempt to discern the different electrical characteristics of nanocapacitors on this sample via the G-IV method.

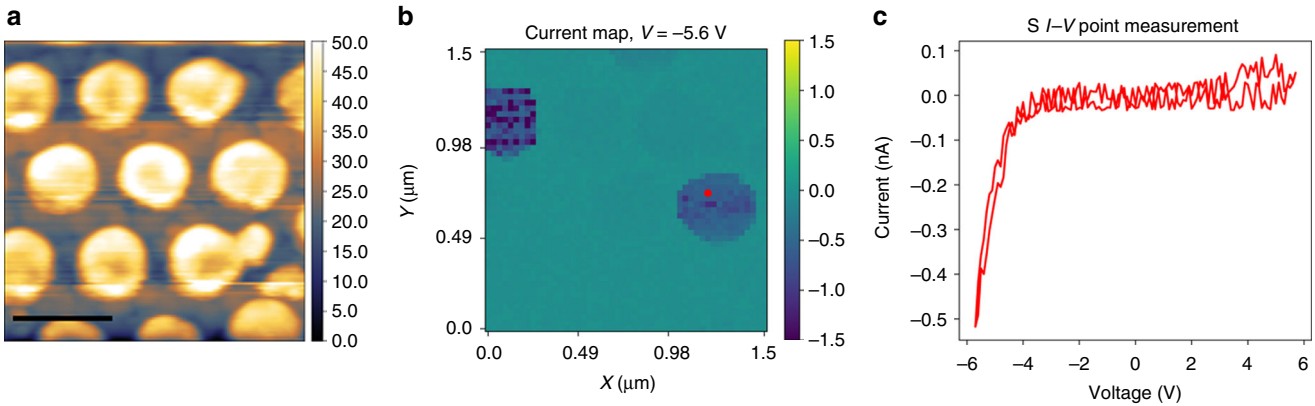

**Fig. 2** S-IV spectroscopy on PZT nanocapacitors. **a** Topography of the region where a spectroscopic *I–V* experiment was performed. Scale bar, 500 nm. The results of the spectroscopy study are shown in **b**, where the current map at *V* = −5.6 V is shown. An example of a single point *I–V* from the region marked by the red circle in **b** is shown in **c**

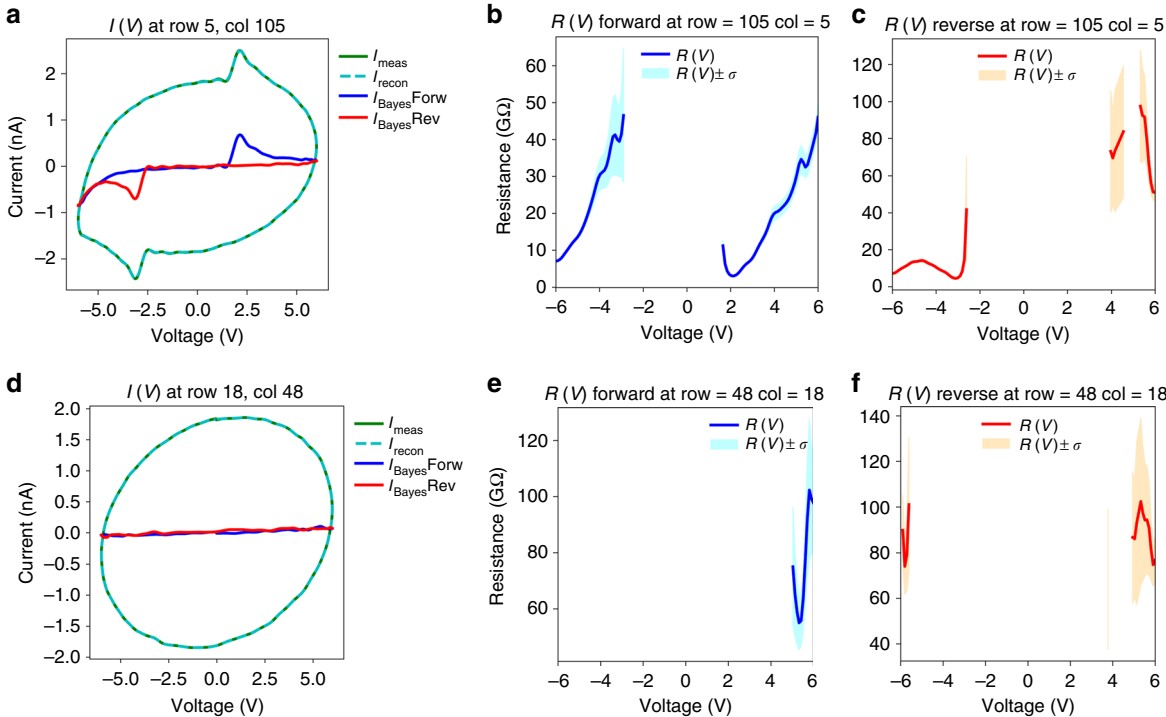

**Fig. 3** Bayesian inference for G-IV. **a**, **d** *I–V* curves measured on the nanocapacitor (**a**) and the bare film (**d**) via the G-IV method. The measured current is plotted as $I_{meas}$, and the current is reconstructed via the Bayesian inference of the capacitance and resistance profiles ($I_{recon}$). The curves without the circuit (capacitance and $a_0C$) contributions are plotted as solid blue and red lines for the forward and reverse traces, respectively. The inferred $R(V)$ profiles are shown in **b**, **c** and **e**, **f** for the two cases (**a**, **d**), for both forward and reverse traces, and are plotted along with their standard deviation (colored regions)

**Standard IV spectroscopy**. The topography of a 1.5 μm × 1.5 μm region is shown in Fig. 2a, and the nanocapacitors and the bare PZT film are both visible in the image. S-IV spectroscopy was then carried out in this region across a 50 × 50 grid of points, resulting in 2500 *I–V* curves in 120 min. A voltage slice of the 3D data set is shown in Fig. 2b, which shows the current at *V* = −5.6 V (see Supplementary Figure 6 for more analysis). Interestingly, it indicates that some of the nanocapacitors display higher conduction than others, due to higher leakage current. A single point measurement from the red circle in Fig. 2b is shown in Fig. 2c, and confirms that the current is mostly on the negative voltage side, and no evidence of a switching current can be seen. For reference, note that the switching voltage for these capacitors

ranges from 2 to 4 V (see ref. [40]). For most points on the film, no current can be detected (i.e., the current is below the noise floor).

With a baseline established from the S-IV measurement, we then proceeded to perform G-IV measurements in the same area. We biased the tip with a 200-Hz AC excitation and captured the sample current through the current amplifier. In total, the G-IV method took 17 min for the complete spectroscopic measurement, yielding >120,000 *I–V* curves, with an increased spatial resolution of 256 × 256 for the same 1.5 μm × 1.5 μm area. To enable processing, we first proceeded to filter the data through FFT filtering techniques (Supplementary Figure 7). A typical *I–V* curve after filtering in this manner is plotted in Fig. 3a, as a solid green line. Due to the large capacitance contribution, the curve is

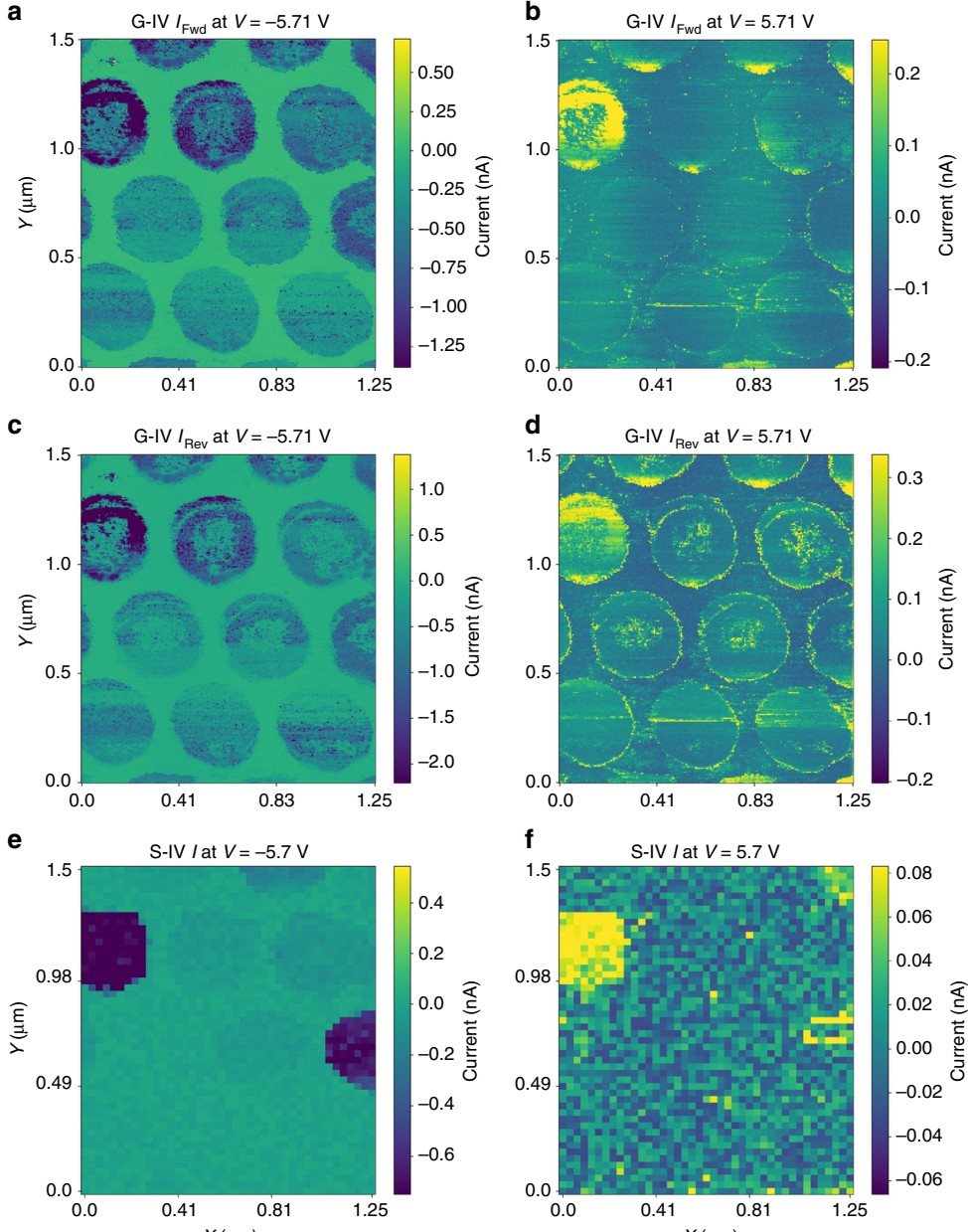

**Fig. 4** Current maps for standard and G-IV methods. After reconstruction of the current at every pixel, it is possible to visualize the results of the G-IV acquisition in the same manner for the standard IV measurement. Shown in **a–d** are voltage slices (taken at both negative and positive extremes) for the forward (**a**, **b**) and reverse (**c**, **d**) voltage directions. The maps of current from the standard IV measurement, in the same region, are shown in **e**, **f** for **e** negative and **f** positive voltages. In general, there is a good agreement; however, exact correspondence is not expected, due to the detection of ferroelectric switching current in the G-IV method (absent in the standard IV measurement)

oval shaped, although kinks associated with ferroelectric switching can be seen at both the positive and negative ends.

**Bayesian inference method**. To invert the dynamic data to yield the local $R(V)$ and the capacitance, $C$, we adapt the Bayesian inference method. That is, we would like to perform fully non-parametric Bayesian inference on the resistance $R(V)$. However, the methods available for this purpose are computationally intensive and make inversion over infinite dimensional function space extremely challenging. By leveraging a linear Gaussian statistical model for an $N$-dimensional approximation of the (inverse of the) function $R$, the inference reduces to identifying mean and covariance, or $N(N+1)/2$ parameters in total (as

opposed to infinitely many in the non-parametric case). Furthermore, these parameters can be identified in closed form through the solution of a linear system. That is, since the parameter to observation map is linear, the mean is given by the solution to a linear system of equations, which is easily computed. Consider the case that $V = V_0 \sin(\omega t)$, and change variables from Eq. (2) with $S = R^{-1}$, so the measured current is

$$I(t) = V_0 \sin(\omega t) S(V_0 \sin(\omega t)) + CV_0(\omega \cos(\omega t) + a_0 \sin(\omega t))$$

(4)

for each $t \in \mathbb{R}_+$, where $\mathbb{R}_+ = \{t \in \mathbb{R}; t \geq 0\}$. Assume that $S$ can be well-approximated by an expansion of basis functions $\phi_i$ so that $S(x) = \sum_{i=1}^{M} s_i \phi_i(x)$, and denote $\bar{s} = [s_1, \dots, s_M]^T$

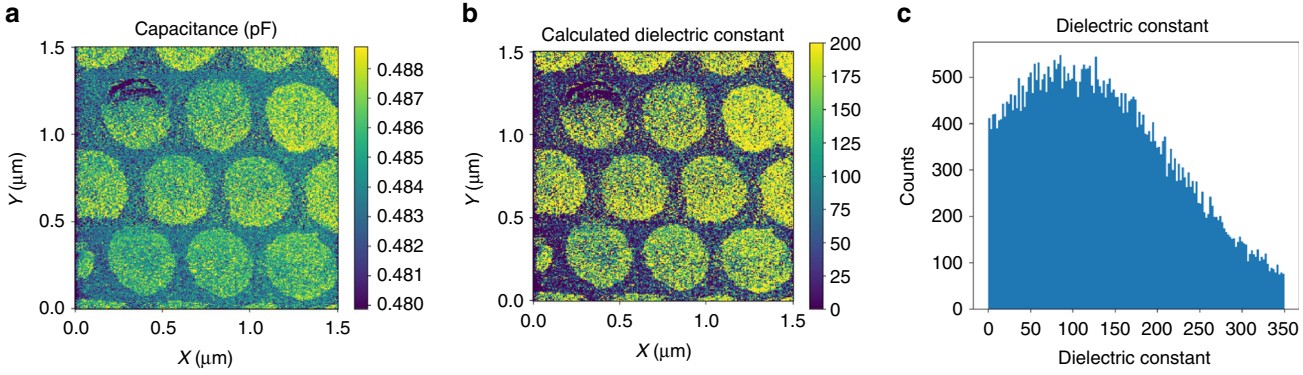

**Fig. 5** Capacitance and dielectric constant determination. **a** Spatial map of the capacitance, derived from the inference method. Subtraction of the measuring system capacitance (taken as the value of $C$ on the bare film) from the capacitance measured on each capacitor allows calculation of the dielectric constant, plotted in **b**, with a histogram of these results plotted in **c**. The histogram peaks at ~100, which is reasonable for this composition of PZT thin film

and $s = [\bar{s}^T, C]^T$. For a given time series $TS = [0, \tau, 2\tau, \ldots, (N-1)\tau]^T$, we define $\overline{A} \in \mathbb{R}^{N \times M}$ to be the matrix such that

$$\overline{A}_{nk} = V_0 \sin(\omega\tau(n-1))\phi_k(V_0 \sin(\omega\tau(n-1))). \quad (5)$$

Define $dV$ as the vector with entries $(dV)_n = V_0\omega\cos(\omega\tau(n-1))$ and $V_n = V_0\sin(\omega\tau(n-1))$. Then Eq. (4) can be evaluated at TS, and can be written as $I(s) = \overline{A}\bar{s} + C(dV + a_0V)$. For convenience, we define the matrix $A$ by $A_{:,1:M} = \overline{A}$ and $A_{:,M+1} = dV + a_0V$. The problem is now to reconstruct $S$ given noisy measurements of the current $I$, i.e.,

$$y = As + \eta, \quad \eta \sim N(b, \Gamma), \eta \perp s, \quad (6)$$

where $N(b, \Gamma)$ denotes a Gaussian random variable with mean $b$ and covariance $\Gamma$. This is achieved by endowing $s$ with a Gaussian prior and using its posterior mean to construct estimators for the resistance $R(V)$ and the capacitance, $C$, and its posterior variance to estimate the uncertainty. Full details of the inversion are given in Supplementary Methods.

The Bayesian inference method has several advantages over previous methods[41–45] that have been proposed to quantify the capacitance contribution (see also Supplementary Note 2 and Supplementary Figures 8,9 for comparison with direct inversion). Specifically, it does not require multiple measurements (e.g., at different frequencies, or reference samples) to determine the magnitude of the capacitive contribution. Moreover, although the method requires that $C$ is constant, it does allow spatial variations of $C$ (each inference is independent). This is critical, given that capacitance can change across the sample, compensation by simple subtraction (e.g., of high resistance areas) is not likely to yield good results in many cases. Lastly, and most importantly, the inference provides the covariance, which allows regions where uncertainty is large in the reconstruction (i.e., when the current is near the noise floor) to be systematically excluded. It should be noted, however, that our method does require us to know the magnitude of $a_0$, which requires a calibration experiment; however, this limitation can be circumvented with the full non-linear inversion, although that brings its own challenges (discussed later).

**Bayesian reconstruction.** The Bayesian reconstructions of two representative $I$–$V$ curves are shown in Fig. 3 (more examples in Supplementary Figure 10). We consider an $I$–$V$ curve taken on

the capacitor (Fig. 3a) as compared to one on the bare PZT film (Fig. 3d). The $I$–$V$ data after filtering are shown as a dark green line ($I_{meas}$), while the Bayesian reconstruction is shown as a dashed cyan line ($I_{recon}$). In both cases, the $I$–$V$ curves are very well modeled by the reconstruction, suggesting the validity of our approach. The reconstruction of $R(V)$ for both cases is plotted in Fig. 3b, c, e, f, for the forward and reverse sweeps of the voltage curve (with forward defined as the sweep from negative to positive voltage, and reverse as the opposite). The limits of the (positive and negative) standard deviations of the reconstruction at each point are plotted as filled colored areas, and increase as the voltage approaches zero, suggesting a decrease of confidence for low currents. This can be rationalized as arising from current values (not from capacitive current) near the noise floor, which can therefore result in an arbitrarily large resistance. Here, we neglected any point where the standard deviation was >40 GOhm, and therefore those regions are not plotted. On the film, almost no measurable current is detected, just as in the S-IV case. Therefore, the resistances inferred have very high uncertainty (>40 GOhm) and thus most are not plotted. At the extreme voltages (near ±6 V) some current is measurable, and the inferred resistance appears to be between 60 and 80 GOhm, suggesting measured currents <100 pA, establishing the noise floor for this technique on our instrument. However, for the case of the measurement on the capacitor, remarkably one can see clearly the presence of the displacement current arising from polarization switching, as seen in the red and blue curves in Fig. 3a. Thus, $R(V)$ is not symmetric, as can be seen in the reconstructions (we additionally provide complete maps of $R(V)$ and $I(V)$ for different voltages in Supplementary Figures 11,12). This is in stark contrast to the S-IV measurement, which was unable to discern the ferroelectric switching current, likely due to the timescales involved. Given that ferroelectric switching in capacitors can occur in the ns–μs range[46,47], it is likely that delay times on the order of ~ms that are present in the S-IV measurement will preclude the detection of the current arising from polarization reversal due to inevitable $1/f$ noise.

**Comparing G-IV and S-IV.** To determine the relationship between the S-IV and G-IV measurements, we plot the current maps (after subtraction of the measurement circuit contributions, i.e., the capacitance and $a_0C$ terms) in Fig. 4a–d for two different voltages, and compare them with the maps taken at the same voltage from the S-IV measurement, plotted in Fig. 4e, f. Interestingly, the high leakage capacitor located in the top-left of the

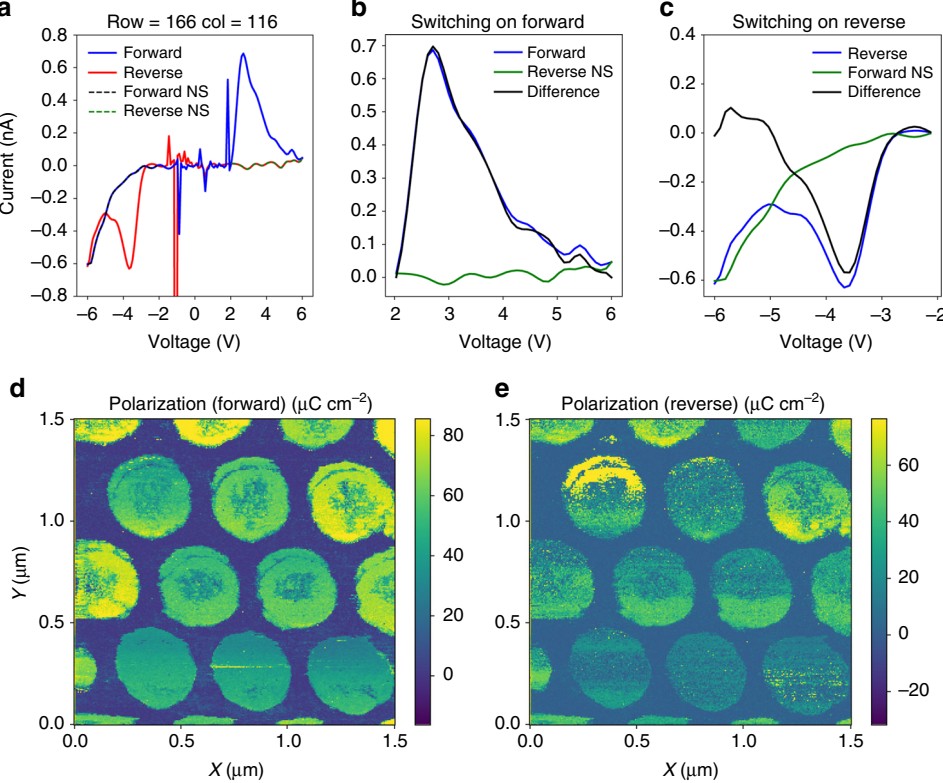

**Fig. 6** Quantitative mapping of polarization. **a** Example of a I–V curve measured on a capacitor (the capacitance contribution has been subtracted). The forward and reverse traces are plotted in blue and red. The non-switching components of the current are plotted as dashed black and green lines, respectively. Subtraction of the non-switching current from the total current allows the switching current traces to be extracted, as shown in **b**, **c** for the forward and reverse sweep directions, respectively. Numerical integration (after conversion from voltage to time, given measurement frequency) allows extraction of the polarization maps, as seen in **d**, **e**. This confirms the quantitative nature and utility of the G-IV technique

image in Fig. 4e, f also stands out in the current maps after the G-IV in the same area (see also Supplementary Figure 13 for I–V point comparison, and Supplementary Figure 14 for analysis of inferred current). However, the agreement between these maps is not exact, and indeed should not be. This is primarily because in this case, the S-IV spectroscopic measurement does not detect the presence of ferroelectric switching current, and therefore comparison with the spatial maps from G-IV is not likely to be successful at all pixels. Indeed, this discrepancy will be more marked when the switching occurs near the end of the voltage window. Nonetheless when compared at points where the switching occurs well within the voltage window, the I−V curves align substantially (Supplementary Figure 13). Furthermore, videos are provided (Supplementary Movies 1 and 2) for the current slices from both methods, and allows clear visualization of the ferroelectric switching contribution in the capacitors from the G-IV measurement, as well as facilitates easy comparison with the S-IV data set.

**Capacitance calculation.** Unlike S-IV, the Bayesian inference also captures the capacitance, allowing us to plot the capacitance variations across the surface, as in Fig. 5a. Unsurprisingly, the capacitors display higher values than the film (capacitance scales with contact area), and again displays why simple subtraction-based capacitance compensation is not feasible for these samples. These results confirm that the G-IV method allows ultrafast IV imaging, and allows determining transient phenomena that are otherwise missed in the standard mode, as well as map the changes in capacitance.

To determine if this capacitance is quantitative, we subtracted the capacitance measured on the film (~480 fF) from those measured on the capacitor, i.e., $C' = C(\text{capacitor}) - C(\text{film})$. From this value, we estimated the dielectric constant, by computing $\varepsilon = \frac{C'd}{A\varepsilon_0}$, where $d = 90$ nm, $A$ is the area of the capacitor (taken as $\pi r^2$ with $r = 250$ nm), and $\varepsilon_0$ is the permittivity of vacuum ($= 8.85 \times 10^{-12}$ F m$^{-1}$). The results of this calculation are plotted as a spatial map in Fig. 5b, and as a histogram, in Fig. 5c, which peaks at around 100 and is consistent with epitaxial PZT films of this composition[48]. Here, we note that the frequencies are very low (sub kHz), and therefore can be considered quasistatic for the ferroelectric; as a result, the derived capacitance and dielectric constants should be compared with the static dielectric constant, and this is in agreement with the literature data for epitaxial PZT films of this composition.

**Polarization inference.** As with standard measurements for testing switching in ferroelectric capacitors, we can obtain further quantitative insight via integration of the area under the switching current to obtain maps of the remnant polarization, as in Fig. 6a–c. The non-switching current, which is obtained on the retrace after switching has occurred, is subtracted from the switching current trace to yield only the switching current contribution, which is plotted in Fig. 6b, c. Subsequently, numerical integration of the curves is performed, and the integration is performed over d$t$ (in the intervals shown in Fig. 6b, c), i.e., we assume that the switching can be considered to be caused by the voltage crossing the threshold voltage for polarization reversal (coercive field), and that the subsequent increase in the voltage

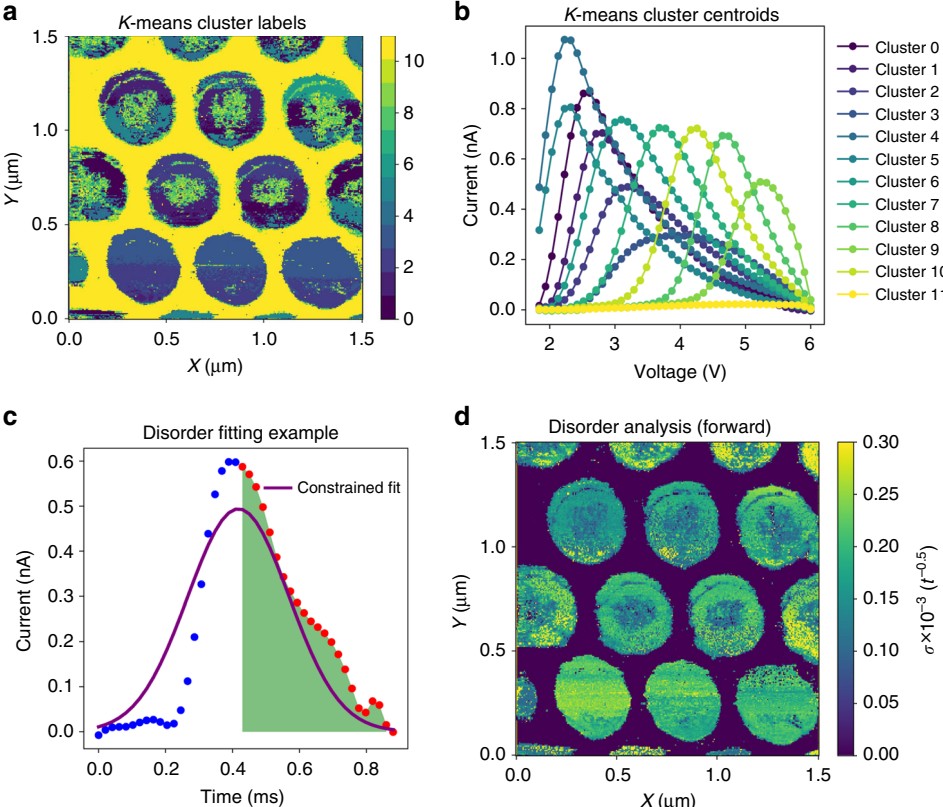

**Fig. 7** Disorder analysis from switching traces. To investigate the variability of the switching current profiles, we performed *K*-means analysis with 12 clusters, restricting the data to the switching current traces in the positive voltage window. The result of the *K*-means analysis is shown in the cluster label map in **a**, and the mean response of each cluster in **b**. Note that clusters with <20 members are not plotted on this graph. To obtain a more quantitative estimate on the disorder, a Gaussian fitting procedure was applied to each switching current profile. An example of the data is shown in Fig. 5. **c** after conversion from voltage to time axis. The procedure involves finding the time at which the integral of the area under the curve is half of the total area under the curve, and fitting to only points at and beyond this time. Such points are indicated as red filled circles in the graph. A Gaussian fit is then performed on these points, with the constraint that the area of the Gaussian be equal to the area of the whole switching curve (to normalize for the polarization). The extracted map of the variance is plotted in **d**, and clearly shows substantial variability from one nanocapacitor to another

does not have an impact. Based on the size of the nanocapacitors and the total collected charge, we can then extract the effective polarization, as shown in Fig. 6d, e, which again agrees well with the expected value for these capacitors[36]. In some nanocapacitors, switching is not observed in the center, likely due to surface contamination in those areas precluding good tip–surface contact. At the same time, the presence of regions of high contrast in Fig. 6e may be indicative of the presence of some irreversible surface electrochemical process in addition to ferroelectric switching and leakage current. A clear imprint in the nanocapacitors is also observed, with the 'down' polarization substantially larger than the 'up' polarization, although the origins are not known.

**Disorder analysis**. More insight into the nature of the ferroelectric switching can then be obtained via analysis of the switching current. Much information can also be obtained via the shape of the switching current trace, as explored in previous works[49,50]. Ferroelectric switching in capacitors is a well-studied phenomenon, with the switching proceeding via nucleation (generally at the electrode[51–53]), and propagation of domain walls to the bottom electrode. For the capacitors in this study, previous work indicates that on an average, there are three nucleation sites per capacitor[40]. Given the material disorder, the nucleation centers will be activated at different voltages, and therefore, the

shape of the switching current trace will reflect this distribution. In the case where no disorder is present, the trace should be a delta function, but strong disorder will cause a longer 'tail' to the switching current curve, as modeled previously by Shur et al.[54]. To determine the types of switching current traces present in the data set, we turned to the *K*-means clustering method (see Supplementary Notes 2 and 9 for more details). The cluster labels map is plotted in Fig. 7a, and the mean response of each cluster is plotted in Fig. 7b (note that clusters with <20 members are not plotted), for the positive voltage region. Interestingly, several of the clusters appear to be characterized by switching current traces with longer tails, indicating greater disorder in these capacitors. To quantify this effect, we follow the procedure outlined in ref. 54 and fit a 1-dimensional (1-D) Gaussian function $y = A_1 \exp\left(-\frac{(x-x_0)^2}{2\sigma^2}\right)$ to the tail of each switching current trace (taking the beginning of the 'tail' as the location where the area under the switching current curve is half of the total area under the curve), with the constraint that the area of the fitted Gaussian equals the area of the whole switching current curve. This allows extracting a value of the variance $\sigma$, which can describe the variance in the switching times in each capacitor. An example of a single point fit is shown in Fig. 7c, with the constrained fit in purple, and the data being fit to indicated in red. The map of $\sigma$ obtained from carrying out this procedure on the whole data set is

shown in Fig. 7d. Thus, this technique allows probing the disorder in nanoscale capacitors, providing a counterpoint to existing and well-developed piezoresponse force spectroscopy on such devices[55].

## Discussion

The use of G-IV and Bayesian inference has allowed for detection and quantitative mapping of the switching response in nanoscale ferroelectric capacitors at unprecedented resolution, at speeds of standard scanning. We note that the Bayesian inference algorithm, although not rapid, has been parallelized and can operate on an entire data set (matrix size $256 \times 256 \times 128$) in under 3 h on a desktop machine. Further improvements and optimizations to the algorithm, as well as integration in high-performance computing environments, should allow the timescale to be reduced dramatically, opening the door to potentially real-time analysis, and paving the way for rapid screening of electronic properties of materials with AFM[56].

Furthermore, the great strength of this technique may lie in its applications toward STM and spectroscopy. Obtaining spatially resolved maps of d$I$/d$V$ is critically important in scanning tunneling spectroscopy, for e.g., exploring quasiparticle scattering[57], metal–insulator transitions[58], molecular state transitions[59], and more, but a major limiting factor is the time taken to obtain atomically resolved maps (often days). The use of G-IV methods in this realm could prove extremely valuable, reducing acquisition times by more than two orders of magnitude and possibly increasing the range of samples on which such experiments can be performed (such as time-sensitive samples, e.g., complex oxide thin films[60]).

The use of the Bayesian inference can also be extended to include further information to constrain the reconstruction and improve accuracy[32]. For instance, extensions include imposing physical constraints directly in the functional form of $R(V)$ and allowing for voltage dependence on $C$, and utilizing hierarchical Bayesian modeling to fit or sample from the (hyper)-parameters which define the prior distribution. Imposing functional form for $R(V)$ requires one to directly invert (as opposed to the economical inversion for $s$). We note that such extensions are potentially able to make better use of the information at hand; however, this improvement comes at a cost of significantly increased complexity, as we will need to resort to expensive Markov chain Monte Carlo (MCMC) methods. These extensions will be left for future study. For AFM, the G-IV method can be applied to investigate a wide variety of electronic phenomena that may have a substantial time-component, including ferroelectric domain wall conductivity, resistive switching and memristive behaviors, and tunneling junctions. By exploring the response as a function of frequency, a frequency-dependent local conductivity map can be extracted, adding a new dimension to existing $I$–$V$ spectroscopy, and which can provide clues to underlying mechanisms[61–63].

## Methods

**Experimental setup**. Details about sample preparation can be found in an earlier manuscript[36]. All experiments were performed on a commercially available microscope (Cypher, Asylum Research). The data were acquired using a National Instruments PXIe 6124 data acquisition (DAQ) card through custom-written LabVIEW-based software instrumentation. S-IV measurements were performed on a grid of spatial locations. G-IV imaging was performed in a conventional two-pass scan mode with force-feedback wherein tip bias was not applied during the trace while the G-IV measurement was conducted during the retrace. During each G-IV measurement, the DAQ applied a sinusoidal excitation bias waveform to the tip and simultaneously acquired the amplified readout from the current amplifier. The DAQ sampled the data at 100 kHz but this rate could be increased if additional time/voltage resolution is desired. The raw two-dimensional (scan row, time) G-IV data was divided to form a three-dimensional data set (row, column, time) such that each spatial pixel contained data corresponding to a single period of the

sinusoidal waveform. Additional information on the spatial/voltage resolution considerations is given below.

**Filtering**. In $I$–$V$ spectroscopy in SPM, the raw current signal is typically noisy since the measured currents are very small (pA to nA). Thus, the conventional $I$–$V$ spectroscopy techniques average the current signal over a few milliseconds. While this approach reduces the noise in the signal, information regarding the dynamics of the system is permanently lost. The availability of the complete and continuous signal in general-mode techniques facilitates data-driven signal processing. Supplementary Figure 7a shows the Fourier transform of the current signal acquired in G-IV for a single line. The noise spectrum shows a clear peak at the excitation frequency (200 Hz in this case) and numerous harmonics of the excitation frequency. The availability of the complete noise spectrum allows the user to attempt different combinations of signal filters after the experiment without having to reconduct the experiment. Further, the data itself can be used to calculate a noise floor instead of using a fixed pre-defined value. Typical G-IV signals are filtered using band-stop filters to remove known noise components that arise due to the instrumentation in addition to a low-pass filter that rejects high-frequency noise. All the signals below the calculated noise floor are also rejected to complete the filtering process. The filtered signal in the FFT space is shown in Supplementary Figure 7b. Finally, an inverse fast Fourier transform returns the filtered signal from the frequency domain to the time domain. Supplementary Figure 7c shows four raw G-IV curves (red curve) as acquired from the current amplifier. The filtered $I$–$V$ curve (blue curve) after data-driven signal filtering shows minimal noise while retaining the finer nuances of the original curve. The necessity for filtering is more pronounced for $I$–$V$ curves obtained from samples with high resistivity.

**Resolution consideration**. For resolution considerations, in the current implementation of G-IV, it is necessary to adjust the SPM scan rate according to the desired spatial resolution and bias frequency. The effective pixel width in G-IV is determined by

$$w_{\text{pixel}} = \frac{w_{\text{scan}} \times 2 \times f_{\text{scan}}}{f_{\text{bias}}}, \tag{7}$$

where $w_{\text{pixel}}$ is the width of the pixels, $w_{\text{scan}}$ is the width of the scan window, $f_{\text{scan}}$ is the scan rate, and $f_{\text{bias}}$ is the frequency of the bias waveform. Thus, if a 50-Hz bias frequency was used instead of 200 Hz for a 200-nm scan window (assuming a scan rate of 0.7 Hz), the width of the pixels would rise from 1.4 to 5.6 nm. Besides being able to investigate, filter, and analyze the complete data, another advantage of general mode is multi-resolution imaging. In other words, the continuous signal captured at each line could be divided into $N$ pixels with $M$ $I$–$V$ curves at each pixel or $N/2$ pixels with $2M$ $I$–$V$ curves within each pixel. So, when the signal-to-noise ratio in the measured signal is low, multiple $I$–$V$ curves could be averaged to get a single curve with lower noise, albeit at the cost of the spatial resolution.

**Data availability**. All the data analysis, processing, and visualization for this paper was facilitated by pycroscopy, which is an open-source python package for analyzing and visualizing microscopy data (see https://github.com/pycroscopy/pycroscopy). Improvements and contributions to pycroscopy are encouraged. The data are available at https://doi.org/10.13139/OLCF/1410993.

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

## Acknowledgements

The work was supported by the U.S. Department of Energy, Office of Science, Materials Sciences and Engineering Division (R.K.V., S.V.K., P.M., S.S.). This research was conducted and partially supported (S.J.) at the Center for Nanophase Materials Sciences, which is a US DOE Office of Science User Facility. The Bayesian inference portion of the research was also sponsored by the Applied Mathematics Division of ASCR, DOE; in particular under the ACUMEN project (K.J.H.L., R.A.). This work was partially supported (Y.K.) by Basic Research Lab. Program through the National Research Foundation of Korea (NRF) funded by the Ministry of Science, ICT & Future Planning (NRF-2014R1A4A1008474). A.N.M. gratefully acknowledges Eugene Eliseev for multiple discussion and technical help, and thanks the National Academy of Sciences, Ukraine, for financial support. M.A. acknowledges the Wolfson Research Merit and Theo Murphy Blue Skies Award of The Royal Society as well as financial support through grant no. EP/P025803/. X.L. acknowledges the financial support of the National Natural Science Foundation of China (Contract no. 51572211).

## Author contributions

S.S. wrote the acquisition software, developed the python analysis codes, conducted the experiments, analyzed the data, and co-wrote the paper. R.K.V. conducted the experiments, analyzed the data, and wrote the paper. K.J.H.L. developed the Bayesian inference method, coded analysis procedures, and co-wrote the paper. R.K.V., S.J., P.M., R.A. and S.V.K. conceived of the idea and co-directed the project. M.A., X.L. and Y.K. were involved with sample synthesis, commented on the analysis, and co-wrote the paper.

A.N.M. developed the analytical and numerical solutions to the series and parallel RC circuits. All authors commented on the manuscript and approved its submission.

## Additional information

**Competing interests:** The authors declare no competing financial interests.

