## [Peer Review File · Nature Communications]

Reviewers' comments:

Reviewer #1 (Remarks to the Author):

This exciting manuscript by S. V. Kalinin, S. Jesse, and R. K. Vasudevan et al. demonstrates a new method of capturing spectroscopic I-V measurements by utilizing the raw current data produced by applying a sinusoidal voltage between the AFM tip and sample. The authors analyze this acquired data by utilizing Bayesian inference to extract the resistive and capacitive behavior of the device that is being tested. This method is performed quickly, allowing them to eliminate potential artifacts introduced by sample drift. To test the validity of the technique, the authors apply standard I-V methods, as well as their newly developed G-IV technique to a model PZT nanocapacitor. Although there are some intriguing difference between the two sets of results that could bear more discussion, there is generally good agreement, and the G-IV method reproduces the expected behavior of these systems, as well as provides insight into the switching behavior that is not discernible with the standard IV spectroscopy technique.

Importantly, the paper represents a well thought out and novel proof of concept for an experimental technique that is likely to inspire many others to attempt the use of Bayesian inversion to extract useful information, and is likely to become highly cited. The technique developed within this manuscript would be of interest to a wide array of scientists, and would allow for more in depth characterization of materials of interest to these researchers. I think addressing the following comments would further increase the impact of this interesting manuscript.

Major issues:

1. In Figure 2a, the topography image shows what appears to be an artifact introduced at the bottom of many/most of the nanocapacitor structures. The authors should discuss this feature, it's origin, and potential impact on the subsequent measurements.

2. I'd really like to see the conventional current maps and reconstructed inverted current maps *aligned* and plotted side by side in the same figure for at least 2 biases, as well as showing conventional and inverted current spectra for the same point. Importantly, as I compare the conventional and inverted current maps, there isn't a 1-1 correspondence. The authors do plot the -5.6V standard map in Fig. 2, and +/- 6V in Fig. 4, but state that tip wear accounts for some of the discrepancy. However, while the inversion is clearly working to some extent, I feel like the discrepancies deserve more comment:

a) the inverted data show higher currents in rings around the edges of the capacitors that correlate with the thicker topography, this trend is absent in the conventional grid, even though the resolution should be sufficient. Can the authors rule out topographic crosstalk during the faster scan?

b) although the text specifies that the conventional and G-IV data were taken on the same region, they clearly appear misaligned by about half a capacitor pad, which makes it inconvenient to compare the image sets, especially with them contained in separate figures

c) not all the "high current" pads in the GIV image appear as high current in the conventional image – and there are pads in between other pads in scan order that don't match up suggesting that simple tip wear probably cannot account for the discrepancy (i.e. the discrepancies aren't emerging as a monotonic function of time but appear and disappear during the scans depending on which pad is being examined)

3. As what is likely to be a highly cited paper, the authors should do a better job explaining the "there's no such thing as a free lunch" (or, in this case, there's no such thing as a free increase in S/N). Emphasizing the points below (and likely others). The applied inversion assumes a linear current/voltage model – and while some samples of interest fall into this category, the majority will exhibit complicated and non-linear waveforms. A frank discussion of the limitations of the technique, and the requirement for knowing a priori what the dI/dV relationship is probably won't stop people in the future, but would at least let this reviewer sleep better. This limitation IS discussed nicely in the

conclusions, but I think it would be best to discuss it explicitly when first discussing the inversion since the introduction talks about more general applications like tunneling spectroscopy and the putting a disclaimer in the conclusion seems to be burying an important point in the fine print. The paper stands on it's on even with this admission so why not own it up front?

Minor issues:

1. Figures 2 and 3 appear to be reversed in order.
2. Labeled scale bars are lacking on all images. It would be helpful to have them on figure 2, 4, 5, 6.
3. On line 130, It is not clear to which noise term the authors are referring, is it to the SI?
4. On line 133, are there better related applications of Bayesian inference? These examples are quite unrelated to the technique being developed.
5. On line 179 the authors make the claim that the G-IV method takes 17 minutes to complete, while directly comparing it to the 120 minutes taken by the standard technique. This comparison is slightly misleading given the fact that the time taken by the Bayesian inversion analysis (3 hours) is not considered, though it is true that this time will be reduced by improvements in computation over time.
6. On line 192, consider an alternate symbol for the time series T . The current choice introduces confusion between the time-series, T , and the transpose operator, T .
7. Dielectric constant, and hence capacitance, is frequency dependent. 7(a) Please specify the frequency(ies) when discussing agreement between reconstructed and expected values (b) mention implications for the analysis of a frequency-dependent capacitance term

Reviewer #2 (Remarks to the Author):

The manuscript "Ultrafast current imaging by Bayesian inversion" describes the development of a much needed method to speed up I-V mapping. The incredible speed-up by three orders of magnitude is offset by complicated and relatively slow data analysis (3 hours). This downside can easily be tackled with optimisation, for example GPGPU programming. It is, however, questionable how much of the true signal is restored.

While some of the data presented is technically impressive the paper is very difficult to follow, and lacks crucial information. The paper, as is made clear from the title, is about how to use Bayesian inversion to perform ultrafast current imaging. To clearly describe such a development one would expect

1. A clear description of the method
2. A direct side by side comparison of the results of the new and old method
3. In depth discussion (qualitative and quantitative) of the differences of the old a new method, and the limitations of the estimated inversion

However to these points:

1. The inversion method is described but not in a way that is clear for a general researcher in the field. The supplementary information does help to improve the clarity.
2. Comparisons of I-V maps is done in separate figures making one to one comparison more difficult. These are however simply slices through a 3D data set, built from an array of I-V curves, as the curves are undergoing reconstruction it is far more important to compare them. The comparison of these is moved to supplementary information and shows poor agreement.

3. There is very little qualitative comparison of data from the two methods, this is generally limited to a couple of I-V maps, which show similar features, which is hardly surprising even for poor inversion. Neither the main text nor the supplementary information properly discusses the clear differences between true I-V curves and the reconstructed curves except to mention tip wear which is very unconvincing. No quantitative comparison is presented and the limitations of the inversion are not discussed. As such the naming of the method "General mode" is rather overselling what has been achieved.

As the paper describes and estimated reconstruction of data, but the quality of the reconstruction and its limitations are not discussed. I suggest the paper should be significantly altered to properly compare the new data with normal I-V spectra, and to accurately comment on the limitations of the new method. As it stands the paper is not of the quality I would expect from Nature Communications. Even with the suggested alterations I am doubtful it would reach this standard.

Reviewers' comments:

Reviewer #1 (Remarks to the Author):

This exciting manuscript by S. V. Kalinin, S. Jesse, and R. K. Vasudevan et al. demonstrates a new method of capturing spectroscopic I-V measurements by utilizing the raw current data produced by applying a sinusoidal voltage between the AFM tip and sample. The authors analyze this acquired data by utilizing Bayesian inference to extract the resistive and capacitive behavior of the device that is being tested. This method is performed quickly, allowing them to eliminate potential artifacts introduced by sample drift. To test the validity of the technique, the authors apply standard I-V methods, as well as their newly developed G-IV technique to a model PZT nanocapacitor. Although there are some intriguing difference between the two sets of results that could bear more discussion, there is generally good agreement, and the G-IV method reproduces the expected behavior of these systems, as well as provides insight into the switching behavior that is not discernible with the standard IV spectroscopy technique.

We thank the referee for their positive appraisal of our manuscript.

Importantly, the paper represents a well thought out and novel proof of concept for an experimental technique that is likely to inspire many others to attempt the use of Bayesian inversion to extract useful information, and is likely to become highly cited. The technique developed within this manuscript would be of interest to a wide array of scientists, and would allow for more in depth characterization of materials of interest to these researchers. I think addressing the following comments would further increase the impact of this interesting manuscript.

We again thank the referee for the appreciation that this work has the potential to be highly useful across domains. We have attempted to respond to all of their queries below

Major issues:

1. In Figure 2a, the topography image shows what appears to be an artifact introduced at the bottom of many/most of the nanocapacitor structures. The authors should discuss this feature, its origin, and potential impact on the subsequent measurements.

The referee makes an important point about the topography in this area. It appears to be nothing more than simple slip of the sample, likely due to external vibration. Beyond affecting the individual line in the topography, it is unlikely to have any effect on subsequent lines.

2. I'd really like to see the conventional current maps and reconstructed inverted current maps *aligned* and plotted side by side in the same figure for at least 2 biases, as well as showing conventional and inverted current spectra for the same point. Importantly, as I compare the conventional and inverted current maps, there isn't a 1-1 correspondence. The authors do plot the -5.6V standard map in Fig. 2, and +/- 6V in Fig. 4, but state that tip wear accounts for some of the discrepancy. However, while the inversion is clearly working to some extent, I feel like the discrepancies deserve more comment:

We thank the referee for this comment. We conducted further calibration experiments by using idealized circuits with known resistances and capacitances, and determined that the inversion procedure can be improved via addition of a small a_0C term, i.e. instead of $I = (V/R) + C dV/dt$, we found better agreement with

$$I = V \left(\frac{1}{R} + a_0 C \right) + C \frac{dV}{dt}$$

This value of a_0 varies depending on cabling, but is typically in the 50-300 $F^{-1} \text{ Ohms}^{-1}$ range. Details of this calibration can be found in the revised supplementary, but is reproduced below in Figure R1.

Figure R1: Calibration test with simple parallel RC circuit with resistor ($R = 1.0 \text{ GOhm}$) and different values of the capacitance C for $C =$ (a) 6.8 pF , (b) 10.0 pF , (c) 32.0 pF and (d) 66.0 pF . The measured current is in black. The pure capacitance contribution to the current ($C \text{ dV/dt}$) is plotted in green. The expected Ohmic contribution to the current is plotted in yellow. This leaves an additional Ohmic contribution, which represents the difference, i.e. $I_{\text{measured}} - I_{\text{capacitance}} - I_{\text{true}}$. The value of this contribution appears to be equal to a constant multiplied by the capacitance – $a_0 C * V$. The value of a_0 across this calibration is steady, between 210-240.

By adding this particular term and then subsequently performing the inference, we were able to obtain plots with much better agreement between the standard IV and G-IV approaches. However, as we explain further below, we do not expect to see direct correlation between the standard and G-IV methods. This is because the standard IV approach is unable to determine the

ferroelectric switching current, which greatly affects the obtained current and resistance maps from the G-IV method.

a) the inverted data show higher currents in rings around the edges of the capacitors that correlate with the thicker topography, this trend is absent in the conventional grid, even though the resolution should be sufficient. Can the authors rule out topographic crosstalk during the faster scan?

This is an interesting point, and we believe these effects do not stem from fast scanning or topographical cross-talk. Instead, we note that the center of the caps does not appear to show much current, and indeed the polarization maps reveal donut-shaped features on many of the capacitors. We suggest the reason for this is the accumulation of adsorbates at the center of the capacitor that prevents electrical contact between the tip and the capacitor. Repeated scanning can clean these areas, and given the standard IV measurement was performed after multiple scans of the same area, it appears a reasonable explanation. At the same time, it may be that scanning the tip makes the pick-up of the adsorbates more likely than in the spectroscopic mode, where the tip is lifted during transit from one point to the next.

b) although the text specifies that the conventional and G-IV data were taken on the same region, they clearly appear misaligned by about half a capacitor pad, which makes it inconvenient to compare the image sets, especially with them contained in separate figures

We have now aligned the images in the manuscript, to facilitate easy comparison. As an example, shown below are the spatial maps for the aligned and registered images between the standard and G-IV methods. Note that because the standard I-V measurement was performed on a 50x50 grid, we have interpolated it to be the same size as the G-IV measurement, so that the images could be registered correctly. It is seen (below in Figure R2) that the capacitor which show high leakage at the upper left in the GIV measurement for both positive and negative voltages shows the same behavior in the standard IV measurement.

Figure R2: Comparison between standard and G-IV methods after the new inference method was performed. (a,b) GIV maps at +5.71 V (a) and -5.71V(b). The aligned (and interpolated) maps from the standard IV measurements at the two voltage extremes for positive (c) and negative (d) voltages are shown.

c) not all the “high current” pads in the GIV image appear as high current in the conventional image – and there are pads in between other pads in scan order that don’t match up suggesting that simple tip wear probably cannot account for the discrepancy (i.e. the discrepancies aren’t emerging as a monotonic function of time but appear and disappear during the scans depending on which pad is being examined)

Here we would like to re-emphasize that we do not expect one-to-one correlation between the two spatial maps shown in Fig. R2. This is because the standard IV does not include current from the ferroelectric switching, whereas G-IV does; this added current contribution will cause some discrepancy between the two maps. Certainly, if one looks at the revised point I-V comparison (plotted below in Fig. R3), we note that the IV curve taken on the capacitor appears to closely follow the reverse current curve from the GIV method, taken on the same capacitor.

Figure R3: Comparison between standard and G-IV methods after the new inference method was performed for a single pixel. Results are shown for a single pixel on the same capacitor with both methods, as well as on the film.

Discrepancies can arise when switching takes place towards the end of the voltage window range. In this case, one would expect to see residual current in the G-IV, but not in the standard IV. As an example, consider the IV curve taken from the point marked by the circle in Fig R4 below. The curve (shown in b) shows the ferroelectric switching current, but this has not completely reduced to zero in the time taken for the experiment. This is a confounding factor in the maps, and we now write this explicitly in the manuscript. Note that it was not possible to extend the voltage window further without damaging the sample or causing excessive wear to the electrical coating of the tip. On page 13 we now write:

“However, the agreement is not exact, and indeed should not be. This is primarily because the standard I-V spectroscopic measurement does not detect the presence of ferroelectric switching current, and therefore comparison with the spatial maps from G-IV is not likely to be successful at all pixels. Indeed, this discrepancy will be more marked when the switching occurs near the

end of the voltage window. Nonetheless when compared at points where the switching occurs well within the voltage window, the I-V curves align substantially (Supplementary S8). Furthermore, videos are given in the supplemental (Supplementary Video 1 and Supplementary Video 2) for the current slices from both methods, and allows clear visualization of the ferroelectric switching contribution in the capacitors from the G-IV measurement, as well as facilitates easy comparison with the standard I-V dataset.”

Figure R4: When switching occurs near edge of voltage window. (a) G-IV current map at $V = -5.71V$. Shown in **(b)** is the plot of the reconstructed current, for the forward and reverse traces from the location marked by the white circle in (a). In this case, taking a voltage slice at $-6V$ will be deceiving, as there is still a substantial contribution from the switching current in the forward trace, and thus cannot be compared directly with the standard I-V measurement.

3. As what is likely to be a highly cited paper, the authors should do a better job explaining the “there’s no such thing as a free lunch” (or, in this case, there’s no such thing as a free increase in S/N). Emphasizing the points below (and likely others). The applied inversion assumes a linear current/voltage model – and while some samples of interest fall into this category, the majority will exhibit complicated and non-linear waveforms. A frank discussion of the limitations of the technique, and the requirement for knowing a priori what the dI/dV relationship is probably won’t stop people in the future, but would at least let this reviewer sleep better. This limitation IS discussed nicely in the conclusions, but I think it would be best to discuss it explicitly when first discussing the inversion since the introduction talks about more general applications like tunneling spectroscopy and the putting a disclaimer in the conclusion seems to be burying an important point in the fine print. The paper stands on it’s on even with this admission so why not own it up front?

We thank the referee for this comment, and modified the introduction section with the inclusion of this paragraph:

“We note here that there is a price to pay for the fast measurements – inherently the faster measurements will suffer from lower signal to noise, and also introduce more uncertainty in in the measurement due to the nature of the inference. However, appropriate de-noising using Fourier filtering appears to work reasonably well, at the cost of very fine features in the current signal (if these are present). At the same time, the nature of the inference allows us to determine the bounds, as we achieve valid statistical estimates of the covariance using this technique. In other words, we trade signal to noise for substantially fast measurements with higher (but known) uncertainty.”

With regards to the linear current-voltage model, $R(V)$ can be decidedly nonlinear (and indeed, it is as shown in our plots in Figure 3 in the manuscript). However, the limitation occurs when the substitution we use can cause problems, i.e. $S = R^{-1}(V)$. This can cause issues because the expectation, $\mathbb{E}(R) = \mathbb{E}\left(\frac{1}{S}\right) = \infty$ as explained in the supplementary material. It becomes necessary for the calculation of the uncertainty; but this problem is circumvented by instead calculating $\frac{1}{\mathbb{E}(S)}$ and $\frac{1}{\mathbb{E}(S) \pm \sigma_S}$. However, this limitation (and indeed, the need to know the constant a_0) are all circumvented by use of full Markov-chain Monte-Carlo approach; nonetheless that is much more computationally expensive, and we leave that for future study.

Minor issues:

1. Figures 2 and 3 appear to be reversed in order.

This has been fixed.

2. Labeled scale bars are lacking on all images. It would be helpful to have them on figure 2, 4, 5, 6.

We have now added scale marks to all image plots in the manuscript.

3. On line 130, It is not clear to which noise term the authors are referring, is it to the SI?

This should just read ‘in the presence of noise’; this has been fixed.

4. On line 133, are there better related applications of Bayesian inference? These examples are quite unrelated to the technique being developed.

We are not aware of any in our domain (of scanning probe microscopy); indeed we believe this fact highlights the cross-disciplinary nature of our work.

5. On line 179 the authors make the claim that the G-IV method takes 17 minutes to complete, while directly comparing it to the 120 minutes taken by the standard technique. This comparison is slightly misleading given the fact that the time taken by the Bayesian inversion analysis (3 hours) is not considered, though it is true that this time will be reduced by improvements in computation over time.

This is a valid point, and we are currently in the process of implementing the features of the inversion in a scalable environment through our open source Pycroscopy package (github.com/pycroscopy/Pycroscopy). In general, this is an “embarrassingly parallel” problem, and it will scale linearly with the number of cores, as each inference is independent. Thus, given access to hundreds of cores, the inference time will be minutes rather than hours. We are currently exploring methods to achieve this.

6. On line 192, consider an alternate symbol for the time series T . The current choice introduces confusion between the time-series, T , and the transpose operator, T .

This is a good point; we have now changed the time-series variable to TS .

7. Dielectric constant, and hence capacitance, is frequency dependent. 7(a) Please specify the frequency(ies) when discussing agreement between reconstructed and expected values (b) mention implications for the analysis of a frequency-dependent capacitance term.

This is also a good point which we did not explain in the initial version of the manuscript. Here we are interested in the static dielectric constant, i.e. that measured at low frequency. The measurement, while much faster than traditional I-V, is still sub kHz, and as such can be considered to be in the quasi-static range as far as the ferroelectric is concerned.

Similarly, the frequency dependence of the capacitance can be taken to be small in this range, but should be a function of the measurement circuit; but we found that in the range of frequencies used for our standard RC circuit tests, there was no measurable dispersion in the frequency dependence of C for the circuit.

Some more discussion on this point has been added on page 13:

“Here, we note that the frequencies are very low (sub kHz), and therefore can be considered quasi-static for the ferroelectric; as a result, the derived capacitance and dielectric constants should be compared with the static dielectric constant, and this is in agreement with the literature data for epitaxial PZT films of this composition.”

Reviewer #2 (Remarks to the Author):

The manuscript "Ultrafast current imaging by Bayesian inversion" describes the development of a much needed method to speed up I-V mapping. The incredible speed-up by three orders of magnitude is offset by complicated and relatively slow data analysis (3 hours). This downside can easily be tackled with optimisation, for example GPGPU programming. It is, however, questionable how much of the true signal is restored.

We thank the referee and attempt to respond to their comments below.

While some of the data presented is technically impressive the paper is very difficult to follow, and lacks crucial information. The paper, as is made clear from the title, is about how to use Bayesian inversion to perform ultrafast current imaging. To clearly describe such a development one would expect

1. A clear description of the method
2. A direct side by side comparison of the results of the new and old method
3. In depth discussion (qualitative and quantitative) of the differences of the old a new method, and the limitations of the estimated inversion

We have taken these comments into consideration, and hope that the improvements will satisfy the referee. We do note that the reason for a lack of in-depth discussion between the two datasets is because they are not directly comparable, as we point out in the response to the first referee.

However to these points:

1. The inversion method is described but not in a way that is clear for a general researcher in the field. The supplementary information does help to improve the clarity.

We have tried to improve this in the revised manuscript. We are essentially just solving a system of linear equations, for $S (=R(V))^{-1}$. By using a Gaussian basis, we are trying to find an approximate form of the solution as a sum of the basis functions. To make this clear, we add the following description below, on page 7:

“We would like to perform fully non-parametric Bayesian inference on the resistance $R(V)$. However, the methods available for this purpose are computationally intensive and make inversion over infinite dimensional function space extremely challenging. By leveraging a linear Gaussian statistical model for an N-dimensional approximation of the (inverse of the) function R , the inference reduces to identifying mean and covariance, or $N(N+1)/2$ parameters in total (as opposed to infinitely many in the non-parametric case). Furthermore, these parameters can be identified in closed form through the solution of a linear system. That is, since the parameter to

observation map is linear, this means the mean is given by the solution to a linear system of equations, which is easily computed.”

2. Comparisons of I-V maps is done in separate figures making one to one comparison more difficult. These are however simply slices through a 3D data set, built from an array of I-V curves, as the curves are undergoing reconstruction it is far more important to compare them. The comparison of these is moved to supplementary information and shows poor agreement.

As explained above, we determined that the circuit model was insufficient, and needed one extra term. We then performed inversion using the new model, and find better agreement. We have also recomputed figures to show the side-by-side comparison, but again as noted above, we do not expect complete agreement due to the presence of ferroelectric switching that is measurable on the capacitors in the GIV but not in the standard IV spectroscopy. We have also registered the image to facilitate easy comparison.

3. There is very little qualitative comparison of data from the two methods, this is generally limited to a couple of I-V maps, which show similar features, which is hardly surprising even for poor inversion. Neither the main text nor the supplementary information properly discusses the clear differences between true I-V curves and the reconstructed curves except to mention tip wear which is very unconvincing. No quantitative comparison is presented and the limitations of the inversion are not discussed. As such the naming of the method "General mode" is rather overselling what has been achieved.

Please see response above. We have improved the inference to take into account the extra term that was offsetting our results, and have also better aligned the standard and G-IV approaches. We now provide complete videos of the current maps at each voltage to facilitate easy comparison.

As the paper describes and estimated reconstruction of data, but the quality of the reconstruction and its limitations are not discussed. I suggest the paper should be significantly altered to properly compare the new data with normal I-V spectra, and to accurately comment on the limitations of the new method. As it stands the paper is not of the quality I would expect from Nature Communications. Even with the suggested alterations I am doubtful it would reach this standard.

We have heavily modified the manuscript with new inversions and new comparisons between the methods, as well as more discussion. We also registered the spatial maps and show videos as a function of voltage, for comparison, as well as a wealth of spectral point comparisons. With the additional details, we trust this manuscript will be much easier to follow, and is suitable for

publication in Nature Communications.

REVIEWERS' COMMENTS:

Reviewer #2 (Remarks to the Author):

The authors have updated their manuscript in line with my comments. I still have some minor reservations about the method itself, but the paper is now of a standard where I believe it should be published and the rest of the community can form their own conclusions.

Response to referee's comments

REVIEWERS' COMMENTS:

Reviewer #2 (Remarks to the Author):

The authors have updated their manuscript in line with my comments. I still have some minor reservations about the method itself, but the paper is now of a standard where I believe it should be published and the rest of the community can form their own conclusions.

We thank the referees for their criticisms and concerns. We hope that through our revision, our points are clearer, and the rearrangement of the paper will assist in making it more understandable.